# Importance of the Androgen Receptor Signaling in Gene Transactivation and Transrepression for Pubertal Maturation of the Testis

**DOI:** 10.3390/cells8080861

**Published:** 2019-08-09

**Authors:** Nadia Y. Edelsztein, Rodolfo A. Rey

**Affiliations:** 1Centro de Investigaciones Endocrinológicas “Dr. César Bergadá” (CEDIE) - CONICET – FEI - División de Endocrinología, Hospital de Niños Ricardo Gutiérrez, Buenos Aires C1425EFD, Argentina; 2Departamento de Biología Celular, Histología, Embriología y Genética, Facultad de Medicina, Universidad de Buenos Aires, Buenos Aires C1121ABG, Argentina

**Keywords:** Sertoli, meiosis, AMH, blood-testis barrier, spermatogenesis

## Abstract

Androgens are key for pubertal development of the mammalian testis, a phenomenon that is tightly linked to Sertoli cell maturation. In this review, we discuss how androgen signaling affects Sertoli cell function and morphology by concomitantly inhibiting some processes and promoting others that contribute jointly to the completion of spermatogenesis. We focus on the molecular mechanisms that underlie anti-Müllerian hormone (AMH) inhibition by androgens at puberty, as well as on the role androgens have on Sertoli cell tight junction formation and maintenance and, consequently, on its effect on proper germ cell differentiation and meiotic onset during spermatogenesis.

## 1. Introduction

At birth, the mammalian testis consists of a series of cords formed by immature Sertoli cells and undifferentiated spermatogonia. At this stage, both cell types proliferate by mitosis. The seminiferous cords are surrounded by peritubular myoid cells and the interstitial compartment, formed by Leydig cells together with developing vasculature and lymph vessels. Similarly to other organs, the testis undergoes a series of changes throughout its development. These morphological and physiological changes are more notorious during the period spanning from birth until puberty, the prepubertal stage. The length of this critical period varies greatly between species. While some groups, like humans and other primates, have a prepubertal period that lasts years, other mammals such as rodents have a much shorter one that lasts around 45 days, e.g., the mouse. Despite the variation in their duration, the key changes that occur during this period are consistent across studied species.

Testicular maturation is intertwined with the maturation of the hypothalamic-pituitary-gonadal (HPG) axis, characterized by the existence of positive and negative feedback loops that ensure proper gonadal development and function. Several hormones are involved in testicular maturation, such as follicle-stimulating hormone (FSH), luteinizing hormone (LH), estrogens and androgens.

Androgens participate in processes as dissimilar as the regulation of Sertoli cell maturation, Sertoli-Sertoli and Sertoli-germ cell junction involved in blood-testis-barrier (BTB) formation and maintenance, germ cell proliferation and differentiation [1,2,3] and spermiation [4]. Androgen action occurs through the androgen receptor (AR), which can act through the classical/genomic or the non-classical/non-genomic pathway [5]. Intuitively, maturation processes occurring in the testes are believed to be the consequence of androgen-induced upregulation of target genes. However, work using high-throughput techniques, like transcriptomic studies based on microarray analyses, clearly indicates that the proportions of androgen up-regulated and down-regulated genes in the testes are similar [6].

In this review, we will focus on the androgen-dependent changes that take place in the mammalian testis around pubertal onset, using both human and mouse as models, with special interest in Sertoli cell maturation and germ cell meiotic entry.

## 2. Sertoli Cell Maturation during Postnatal Development

Immature Sertoli cells are the main component of the prepubertal testis. They proliferate actively during the early postnatal period in response to FSH [7,8,9,10,11,12] and other growth factors [13,14,15]. The total number of Sertoli cells that is generated during this stage will have a direct effect on sperm production in adult life, since each Sertoli cell is capable of supporting a certain, fixed number of developing germ cells [7,16,17,18,19].

Immediately after birth, Sertoli cells are small and oval. Their size increases during the prepubertal period due to expansion of their cytoplasmatic volume [20,21]. Concomitantly, Sertoli cells begin to form hemidesmosome-like unions between their basal region and the basal lamina of the tubule [22]. These intercellular unions will ensure the scaffolding of the seminiferous epithelium, which will then support germ cell development throughout spermatogenesis. Therefore, the morphological changes that Sertoli cells undergo as part of their maturation process reflect the changes that germ cells in direct contact with them undergo as well. A key player in this mutual maturation process is the Sertoli cell cytoskeleton, mainly formed by microtubules, actin filaments and vimentin intermediate filaments [23,24,25,26,27].

As for their physiology, prepubertal Sertoli cells produce high levels of anti-Müllerian hormone (AMH), even in the absence of FSH, and begin to express the AR. In humans, the AR is expressed in Sertoli cells at around 12 months after birth [28,29], whereas in the mouse, Sertoli cells begin to express the AR between 4–5 days after birth [8,30]. Both the number of AR positive Sertoli cells and the expression levels increase progressively until pubertal onset, when all Sertoli cells express the AR. High expression levels of AMH are a trademark of immature Sertoli cells during the prenatal period and prepuberty. By the time puberty begins, AMH levels start to decline as a direct consequence of androgen action on Sertoli cells. We will expand on the evidence available on AMH inhibition in response to androgens in upcoming sections of this review.

Many of the nurse-like and scaffolding roles fulfilled by Sertoli cells are a direct result of the maturation process they undergo from birth to puberty in the mammalian testis. Amongst these changes, the appearance and maintenance of the BTB is of critical importance, since it allows for the creation of two distinct compartments within the seminiferous epithelium and also supports germ cell migration from the basal lamina towards the lumen of the seminiferous tubules [31]. The formation of the BTB is regulated by several hormones, such as FSH and androgens, cytokines and by the presence of the germ cells themselves [32].

### Hormonal Regulation of Sertoli Cell Maturation in the Postnatal Testis

After birth, Leydig cells in the interstitial compartment of the testis continue to produce androgens in response to LH, while FSH induces an increase in Sertoli cell proliferation and AMH production [8,10,33]. The high AMH production and the lack of Sertoli cell morphological changes, typical of maturation occurring at this stage when testosterone production is high, reflect a transient, physiological insensitivity to androgens of the Sertoli cell (Figure 1) [8,28]. Shortly after, e.g., by the 6th month in the human male, the HPG axis enters a quiescent period, which results in a decay in FSH and LH levels. This ‘turning-off’ of the HPG axis leads to the disappearance of functional Leydig cells and, therefore, causes a dramatic drop in androgen production. Concomitantly, FSH decay results in cessation of Sertoli cell proliferation. Nevertheless, immature Sertoli cells continue to produce high levels of AMH, which resembles the gonadotrophin-free context production of this hormone occurring in the fetal gonad [34]. AMH production is a characteristic of immature Sertoli cells, and serum AMH is actually used in patients as a biomarker of prepubertal Sertoli cell function [35,36,37,38,39,40,41,42,43,44]. Interestingly, the androgen-induced decline in AMH expression during pubertal maturation is partially reversed by the depletion of intratesticular androgen concentration provoked by treatment with a gonadotrophin releasing hormone (GnRH) analogue in adult males with prostate cancer [45]. Additionally, low AMH is also a biomarker of the impaired functional status of Sertoli cells in congenital disorders, like gonadal dysgenesis [46], or acquired conditions, like in chemotherapy-induced testicular toxicity [47].

As previously mentioned, prepubertal immature Sertoli cells begin to express the AR at 12 months-old in humans [28] and around 4–5 days after birth in the mouse [8,30]. The increase in AR expression happens in a testosterone-free environment, thus not inducing Sertoli cell maturation. If due to abnormal conditions, testosterone production is maintained during the expectedly “quiescent” period, AMH expression is inhibited, reflecting precocious Sertoli cell maturation [48,49]. 

The importance of the AR signaling pathway in the male reproductive system has been studied in depth. The promoter for the AR gene lacks a typical TATA box and, in agreement with many TATA-less genes, transcription is driven primarily by binding of the zinc finger transcription factor Specificity Protein 1 (Sp1) to GC box regulatory elements [50]. In human prostate cancer LNCaP cells, inhibition of Sp1 activity results in a strong decrease in the AR protein level [51], showcasing Sp1 relevance in the regulation of AR transcription. However, the factors that trigger AR expression, particularly in the Sertoli cell, remain yet to be determined.

Reactivation of the HPG axis at puberty results in the reappearance of Leydig cells [52,53,54], which are now active and start producing androgens in increasing amounts. This strong intratesticular androgen production is maintained throughout puberty and adulthood. At the onset of puberty, Sertoli cells already show a strong expression of the AR and are, therefore, sensitive to androgen action (Figure 1), which brings about a decline in AMH production as a result of a direct action on Sertoli cells [30,40,55,56,57,58,59]. As a consequence of the Sertoli cell maturation process BTB formation commences [23,24,25,26,27]. Concomitantly, germ cells enter meiosis and sperm production ensues.

## 3. AR Signaling in Sertoli Cells

The AR is a member of the ligand-activated nuclear receptor superfamily, which includes receptors for estrogens, progestins, glucocorticoids, mineralocorticoids, vitamin D, thyroid hormones and retinoic acid. The AR is encoded by a single copy gene in the X chromosome that is composed by 8 exons [60,61]. The exon-intron boundaries for this gene are conserved in other steroid receptors, suggesting a common ancestor. The classical nuclear/cytoplasmic AR is a modular protein that consists of three functional domains: an N-terminal domain (NTD), a DNA-binding domain (DBD) and a ligand-binding domain (LBD) [62,63]. Androgens act through two different mechanisms: the classical/genomic pathway and the non-classical/non-genomic pathway (Figure 2).

### 3.1. Classical Pathway of Androgen Action

The classical (or genomic) pathway of androgen action involves the nuclear/cytoplasmic AR (Figure 2). Monomers of this receptor are bound to cytoplasmic heat-shock proteins. Binding between androgens and the AR induce conformational changes that result in the release of these monomers from the heat-shock proteins, an increase in receptor phosphorylation, and homodimer formation and interaction with DNA [62,63,68]. The ligand-bound AR dimer can then interact with specific DNA sequences present within the regulatory regions of its target genes, known as androgen response elements (AREs). AREs are usually formed by two palindromic regions 5′-AGAACA-3′ joined by a 3-non-defined-base spacer, with the human consensus ARE being 5′-AGAACAnnnTGTTCT-3′ [69,70]. There are both consensus and non-consensus ARE sequences that have been described for known androgen-regulated genes such as *Rhox5* [71], *Cyp17* [72], *Eppin* [73] and *Tubb3* [74]. This is a relatively slow mechanism, requiring 30 to 45 min for transcriptional regulation after androgen stimulation, and additional time is required for the response to be reflected at the protein level [75].

Although recent microarray studies have identified similar numbers of up-regulated and down-regulated genes in Sertoli cells during the process of postnatal maturation [76], and especially in response to androgens in Sertoli cells [6,77,78], most of the androgen-regulated genes thoroughly studied so far are positively regulated by androgens. Amongst those, *Rhox5* (reproductive homeobox-5), formerly known as *Pem*, is perhaps one of the best characterized androgen-responsive genes [79]. *Rhox5* is expressed in prepubertal and pubertal Sertoli cells and its regulation has been studied in detail. This gene has two regulatory regions; a distal region that is independent of androgen action and a region within intron 2 that is androgen-dependent and responsible for its expression in both testis and epididymis [80,81]. Within the intronic regulatory region, there are two AREs that act synergistically and respond in an androgen-specific manner [71].

The ligand-bound AR can also act indirectly by interacting with other trans-activating factors that are bound to the regulatory regions of their target genes, as is the case for the LH subunits α [82] and β [83] genes. This means that AR action is not determined by the presence of ARE sequences. Regardless of the type of interaction between the AR and its target genes, the outcome can be either positive or negative, meaning that androgens can both stimulate or inhibit the expression of their target genes.

### 3.2. Non-Classical Pathways of Androgen Action

The non-genomic (or non-classical) pathway translates signals into changes in cellular function very rapidly, within second to minutes (Figure 2) [5,84,85,86]. In the Sertoli cell, testosterone stimulation provokes the classic AR to localize near the plasma membrane, where it activates Src tyrosine kinase resulting in phosphorylation of the epidermal growth factor receptor (EGFR). Consequently, the MAP kinase cascade is triggered, including the kinases Raf, MEK and ERK followed by the activation of the p90Rsk kinase, resulting in the phosphorylation of target protein, e.g., the transcription factor cyclic-AMP response element binding-protein (CREB)**.**

An alternative pathway, involving a membrane AR, has been described in different cell types [87,88]. Recently, a member of the ZIP zinc transporter family, ZIP9 has been reported as a membrane AR, unrelated to the classic intracellular AR [89]. There is only one report to date in which the role for ZIP9 is shown in Sertoli cells [90].

### 3.3. Co-Repressors and Co-Activators of AR in Sertoli Cells

The AR can interact with a diverse range of proteins, including components of the general transcription machinery, specific transcription factors and proteins that act as co-activators or co-repressors, also known as co-regulators of AR function. The histone acetyltransferase binding to the origin recognition complex, HBO1 (also known as MYST2 in rodents or KAT7 in humans) has been shown to act as a co-repressor of the AR in prepubertal Sertoli cells [91]. HBO1 prevents the action of steroid receptor coactivator 2 (SRC2, formerly known as TIF2), an AR co-activator that interacts with the activation function 1 (AF1) and 2 (AF2) domains of the AR [92]. SRC2 is also involved in cell adhesion between Sertoli cells and germ cells in the adult mouse testis [93,94]. More recently, the orphan nuclear receptor DAX1, encoded by *Nr0b1*, has also been described to act as a co-repressor of the AR in Sertoli cells where it inhibits the expression of ubiquitin-conjugating enzyme E2B (UBE2B) [95].

## 4. Androgens and the Sertoli Cell

As already mentioned, androgen action on Sertoli cells is critical for proper testicular maturation and normal spermatogenesis progression. When the AR is specifically absent from Sertoli cells or it malfunctions, Sertoli cells remain immature, and spermatogenesis is blunted since meiosis does not occur, resulting in infertility. Evidence for these phenotypic characteristics stems from both human [46,68,96,97,98] and experimental mouse models [8,99,100,101,102,103].

Sertoli cell maturation in response to androgens involves both upregulation and inhibition of different genes. We will discuss some examples known up to date that show the stimulatory effect of androgens on several BTB tight junction components in Sertoli cells and on meiotic onset in the pubertal testis. We will also expand on the inhibitory effect of androgens on the expression of a key immaturity Sertoli cell marker, AMH.

### 4.1. Stimulatory Effects of Androgens on BTB-Related Gene Expression in Sertoli Cells and its Role on Meiotic Onset in the Testis

The BTB appears at a time when serum gonadotrophins, FSH and LH, are elevated as a result of pubertal reactivation of the HPG axis [104,105]. While FSH acts directly on the Sertoli cells through its own receptor, LH induces androgen production by the Leydig cells. Androgens act then on Sertoli cells to promote their maturational changes.

The BTB divides the seminiferous tubules into two compartments, basal and adluminal, thus creating two distinct microenvironments. The BTB is both a tight [106] and dynamic structure that keeps separate compartments within the seminiferous epithelium while allowing for germ cell transit from basal to adluminal space during spermatogenesis [26,31,107]. The mature, fully-formed BTB consists of tight junctions, a testis-specific type of adherent junction known as basal ectoplasmic specializations [22,108], gap junctions and desmosomes [105,107,109,110,111] (Figure 3).

Tight junctions are formed by claudins, namely claudin-3 (CLDN3) and claudin-11 (CLDN11) in the mouse [112,113,114]. Tight junctions interact with the cytoskeleton of Sertoli cells through scaffolding proteins, like Tight junction protein 1 (TJP1, also known as zonula occludens 1 or ZO1) [115] (Figure 3). *Cldn3*, *Cldn11* and *Tjp1* are all expressed throughout postnatal development in the mouse testis [116,117,118] and their proteins localize to the BTB region from pubertal onset onwards [117,119,120]. In mice, the expression of *Cldn11* and *Tjp1* increases progressively from birth, with a marked increase around day 10—in coincidence with the upsurge of first meiotic division—and remains elevated throughout adulthood [121].

In the gonadotrophin-deficient hypogonadal (*hpg*) mouse, spermatogenesis is arrested at the prepubertal stage when meiosis has not begun yet, in association with a disorganization of the tight junctions resulting in the lack of a properly formed BTB. This phenotype stems from the lack of maturation of the Sertoli cells in the absence of androgen production due to a disrupted HPG axis [99,122,123]. In the tubules of *hpg* mice there is no CLDN3 expression and CLDN11 is localized to adluminal areas of Sertoli cells. When treated with FSH alone, *hpg* mice recovered normal CLDN11 distribution, but the tight junctions were still unable to function as a proper barrier. In contrast, treatment with DHT induced a normal distribution of CLDN11 and an increase in the expression of both *Cldn3* and *Cldn11* genes [124].

Evidence of androgen-dependency of the BTB for its appearance and maintenance also derives from studies in mice lacking proper AR expression or function. While general defects in BTB formation were initially described in *Tfm* mice [125], mouse models that either lack AR expression completely (ARKO mice, [101]) or in Sertoli cells only (SCARKO mice, [102,118]) have provided evidence for many genes potentially involved in BTB formation around pubertal onset and maintenance through puberty and adulthood. Histological and electron microscopy studies showed a clear disruption of the BTB in SCARKO mice [118], and the use of microarrays allowed for the identification of androgen-regulated genes involved in BTB formation [126,127].

The expression of *Ocln* (Occludin) and *Cldn11* is inhibited in the absence of androgen action as seen in SCARKO mice [118,128,129,130], and the same occurs with *Tjp1* [131] and *Cldn3* [118]. While FSH plays a role in the regulation of *Cldn11* expression to a lesser extent than androgens [121], this is not the case for *Tjp1*, which is strongly inhibited in SCARKO mice but not in FSHRKO mice [131]. Changes in gene expression have been shown with a classic RT-qPCR approach [118,121,130] and also with RNA-Seq [127].

Another example is that of *Claudin-13* (*Cldn13*) and a non-canonical *Tight junction protein 2 isoform* (*Tjp2iso3*), which have been shown to be downregulated in the SCARKO^tm2.1^ model [132]. Both *Cldn13* and *Tjp2iso3* have several putative ARE sequences, mainly with the TGTTCT motif, to which the AR can bind, as seen by ChIP-qPCR. While CLDN13 is part of the Sertoli cell tight junction, TJP2iso3 participates in tricellular junctions. Furthermore, new candidate genes associated with cell-adhesion and cytoskeleton dynamics show altered expression levels in the SCARKO mouse testis, such as *Actn3* (actinin-a3), *Ank3* (ankyrin 3), *Anxa9* (annexin A9) and *Scin* (scinderin) [118]. However, their involvement in BTB integrity remains yet to be unveiled and much remains to be investigated.

Recently it has been shown that dehydroepiandrosterone sulfate (DHEAS) stimulates the expression of *Cldn3* and *Cldn5* in the mouse Sertoli cell line TM4 through a membrane-bound G-protein-coupled receptor that interacts with Gnα11 and induces phosphorylation of ERK1/2, CREB and ATF1 [133]. This mechanism would mimic the non-classical/non-genomic pathway of androgen action.

Coincidentally with the disorganization and delay in BTB formation, there is an incomplete meiosis in the testis of both *Tfm* and SCARKO mice. The lack of complete meiosis progression in the absence of the AR, specifically in Sertoli cells, demonstrates the central role that androgen-signaling through Sertoli cells plays on spermatocyte entry into meiosis [102,103]. The dynamic nature of the BTB is fundamental for migration of meiotic germ cells from the basal to the adluminal compartment.

On the other hand, in a transgenic mouse model with Sertoli cell-specific premature postnatal AR expression [134], *Rhox5* levels were elevated. Furthermore, there was a precocious upregulation of tight junction markers *Cldn11* and *Tjp1* resulting in early BTB and seminiferous tubular lumen formation, associated with premature meiotic onset, shown by increased levels of meiotic markers *Dmc1* (DNA meiotic recombinase 1) and *Spo11* (SPO11 meiotic protein covalently bound to double strand breaks).

The connection between androgen-induced Sertoli cell maturation and germ cell entry into meiosis remains yet to be fully elucidated. A plausible connection could be that the androgen-induced cytoskeletal changes within Sertoli cells might cause changes in the germ cell cytoskeleton itself, thus promoting transition into meiosis and germ cell movement through the BTB. A crucial role for Sertoli cells in the establishment of an immunoprivileged microenvironment at the time of tight junction formation has also been suggested [117,135]. Whether any of these are the case or not, it is clear that androgen action through the AR on Sertoli cells is pivotal to initiation of meiosis in the pubertal testis, since when the AR is absent, there is no complete meiotic progression.

### 4.2. Inhibitory Effect of Androgens on AMH Gene Expression in Sertoli Cells

Inhibitory effects of androgens on gene expression have not been as extensively studied as the stimulatory ones, with few examples available to date. Genes coding for WNT5A and podoplanin are down-regulated through unknown molecular mechanisms [59]. The androgen-inhibited genes through AR binding to ARE include *Maspin* [136,137] and *Ccnd1* [138]. Representing inhibited genes without functional ARE on their promoter regions that rely on AR interaction with trans-activating factors are *Ngfr* (Nerve growth factor receptor, formerly Neurotrophin receptor p75) [139] and the genes encoding the α- [82,140] and β- [83,141] subunits of LH.

As previously mentioned, the decrease in AMH expression at pubertal onset is indicative of Sertoli cell maturation. Despite the fact that AMH downregulation by androgens has been established a long time ago in all animals studied, including human [55,58,96,142], mouse [8,48,49,143], ram [144,145], pig [146,147], stallion [148], bovine [149,150], and tammar wallaby [151], it has not been until recently that the underlying mechanism of androgen action was described [30]. 

Sertoli cells begin to express AMH early during gonadal development, at 7 weeks in the human embryonic gonad [34] and at 12.5 days post-*coitum* in the mouse male gonad [152]. The expression of the *AMH* gene relies on the presence and action of several transcription factors that bind to their promoter, namely SOX9, SF1, WT1, GATA4, AP2 and NFκB [143]. AMH transcription is dependent mainly upon SOX9 binding to the promoter, but it also relies on SF1 action. SF1 can bind directly to the *AMH* promoter and also interact with other transactivating factors, such as SOX8, to induce AMH expression (Figure 4A). When SF1 is absent, AMH expression drops dramatically [153]. When interaction between SF1 and SOX8, SF1 and WT1 and/or SF1, SOX9 and GATA4 is disrupted by interaction of DAX1 with SF1, AMH expression is inhibited in Sertoli cells [154,155]. This inhibitory capacity of DAX1 on AMH, however, has no relation to androgen action, since it has been described at a time when Sertoli cells do not express the AR and are, therefore, insensitive to this type of hormones. At pubertal onset, the androgens testosterone and dihydrotestosterone have a direct negative effect on *AMH* promoter activity in Sertoli cells. This inhibitory effect involves the proximal region of the *AMH* promoter and requires the presence of the AR together with at least one intact binding site for SF1 in the promoter of the *AMH* gene [30]. These findings were shown using a mouse prepubertal Sertoli cell line [156] and suggest that the inhibitory effect of androgens on AMH expression could be due to direct interaction between the AR and SF1 or by the AR blocking SF1 binding sites, thus preventing SF1 from exerting its stimulatory action on the *AMH* promoter (Figure 4B) [30]. A similar mechanism of action posing an interaction between the AR and SF1 has been described for the androgen-dependent inhibition of the LH β subunit gene [141].

Inhibitory action of androgens on gene expression at puberty in the male is not as common as stimulatory effects. AMH is an immaturity Sertoli cell marker that is regulated by androgens in a negative manner, thus presenting itself as a clear example for androgen inhibition.

## 5. Concluding Remarks

Sertoli cells constitute the physical and physiological foundation of the seminiferous epithelium. They are the link between the HPG axis and germ cells and, therefore, sperm production. To ensure their many roles in the adult testis, Sertoli cells must mature in a timely manner and they do so by preparing themselves to respond to androgen action at the right time. Androgens are responsible for the occurrence of several pubertal development-related events in the testis, most of which are known to be dependent on the stimulatory role of androgens.

Immature Sertoli cells are impervious to androgen action because they lack AR expression. Once the AR becomes present in Sertoli cells and androgen levels increase at pubertal onset, a consortium of genes—like tight junction-associated genes involved in the formation of the BTB—is upregulated, while others—like AMH—become repressed, together depicting the androgen-dependent process of Sertoli cell pubertal maturation. As a consequence of androgen action, Sertoli cell maturation sets a favorable environment for germ cell entry to meiosis and the full progression of spermatogenesis.

## Figures and Tables

**Figure 1 cells-08-00861-f001:**
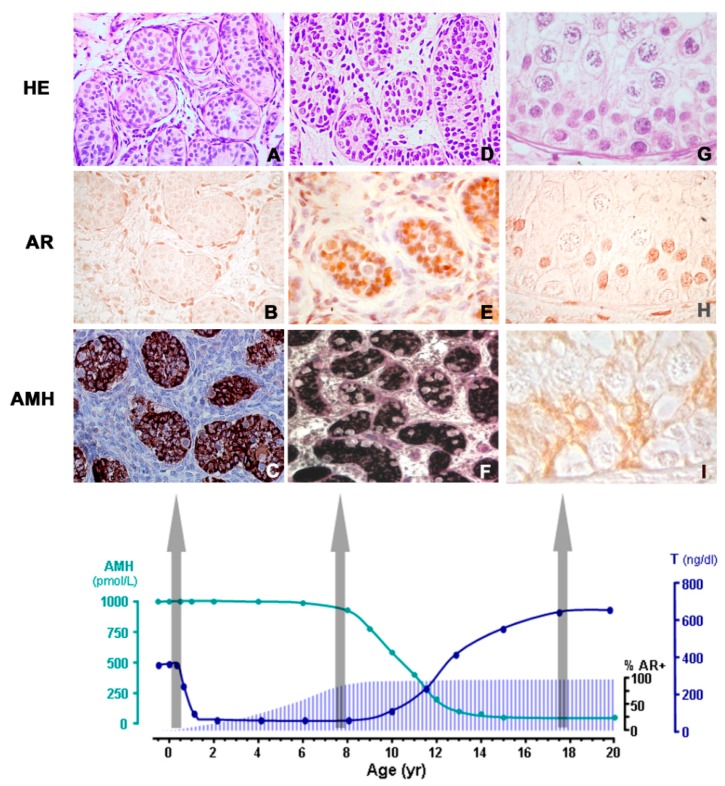
Androgen levels, androgen receptor (AR) expression and AMH in the human testis from fetal life to puberty. **A**–**C**: During infancy, testosterone levels are high, but they do not induce Sertoli cell maturation because the latter do not express the AR: AMH is high, and germ cells do not enter meiosis. **D**–**F**: During the “quiescent” period of the hypothalamic-pituitary-gonadal axis occurring in childhood, most Sertoli cells express the AR (immunohistochemistry), but there are no mature Leydig cells in the interstitial compartment and testosterone is low; therefore, Sertoli cells remain immature. **G**–**I**: In puberty and adulthood, the increase in testosterone provokes Sertoli maturation, reflected in the decline of AMH expression, and also in the onset of adult spermatogenesis. HE: hematoxylin-eosin stain; % AR+: percentage of Sertoli cells with positive immunostaining for the AR. AMH (pmol/l) and T (ng/dl) reflect schematic AMH and testosterone serum levels from birth to 20 years of age in the human male. Reproduced with permission from Rey et al. 2009 [29]. Copyright 2009, Wiley-Liss, Inc.

**Figure 2 cells-08-00861-f002:**
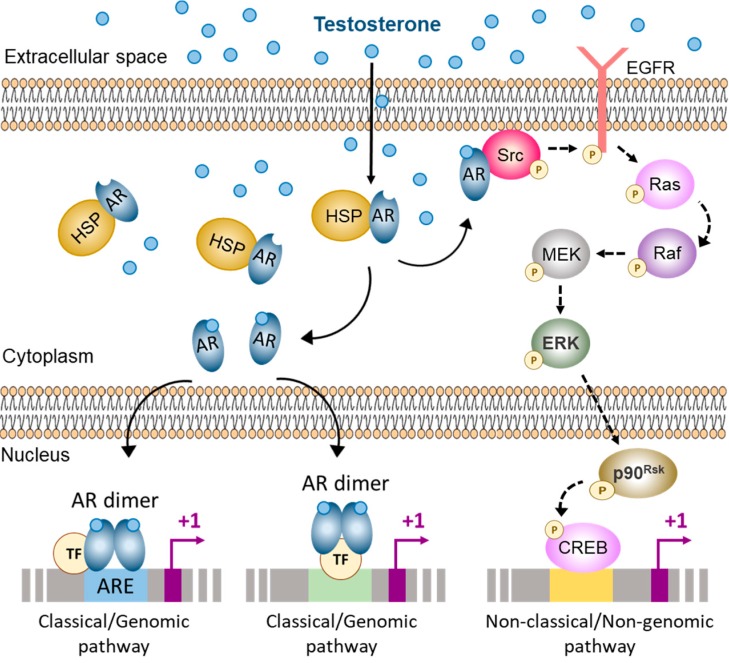
Pathways of androgen action in the Sertoli cell. The classical and non-classical pathways of androgen action co-exist in the Sertoli cell. In the cytoplasm the androgen receptor (AR) is bound to heat-shock proteins (HSP). When androgens bind to the AR it causes a conformational change that releases the AR as monomers. In the Sertoli cell, ligand-bound AR monomers can either migrate to the inner side of the cell membrane and interact with Src, thus activating the non-classical/non-genomic pathway of androgen action; or they can translocate to the nucleus and form homodimers that can interact with androgen response elements (ARE) or with other transcription factors (TF), thus activating the classical/genomic pathway of androgen action. Src: Steroid receptor coactivator, EGFR: Epidermal growth factor receptor, MEK: Mitogen-activated protein kinase, ERK: Extracellular signal-regulated kinase, CREB: cAMP response element binding protein. Based on refs. [5,64,65,66,67].

**Figure 3 cells-08-00861-f003:**
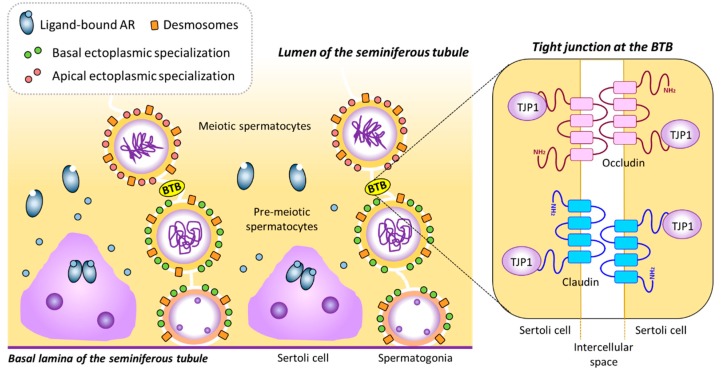
The blood-testis barrier. The BTB is formed by intercellular unions between adjacent Sertoli cells. In the presence of androgens, AR-expressing Sertoli cells can mature and express several genes needed for BTB formation, such as *Cldn3*, *Cldn11*, *Ocln* and *Tjp1*. CLDN3, CLDN11, OCLN and TJP1, together with other proteins and components of the cytoskeleton, such as actin bundles, constitute tight junctions at the BTB. BTB: Blood-testis barrier, TJP1: Tight junction protein 1. Based on refs. [32,107].

**Figure 4 cells-08-00861-f004:**
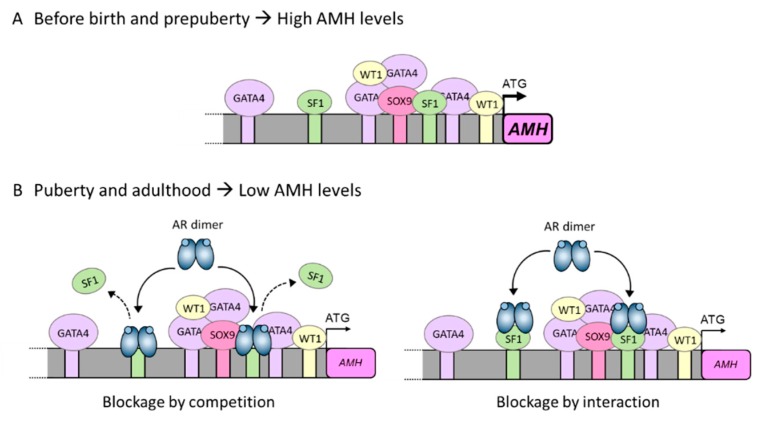
Underlying mechanism for AMH inhibition by androgens in the pubertal Sertoli cell. (**A**) Sertoli cells produce high levels of AMH during the prenatal and prepubertal period. This high expression is a direct consequence of SF1, GATA4 and WT1 interaction with their own binding sites on the *AMH* promoter and also of protein-protein interactions with each other. (**B**) At pubertal onset, Sertoli cells express the androgen receptor (AR) and can respond to androgen action. Androgens inhibit *AMH* promoter activity through the AR. Inhibition could be due either to a direct interaction between the ligand-bound AR and the SF1 sites present on the *AMH* promoter (blockage by competition), or due to a protein-protein interaction between the ligand-bound AR and promoter-bound SF1 (blockage by interaction). In either scenario, the ligand-bound AR prevents SF1 from exerting its stimulatory effect on *AMH* promoter activity, thus resulting in a decrease in AMH expression. Based on ref. [30].

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
