# Peer review of "Importance of the Androgen Receptor Signaling in Gene Transactivation and Transrepression for Pubertal Maturation of the Testis"

_cells, 2019, doi:10.3390/cells8080861_

Round 1

Reviewer 1 Report

The autors reviewed the importance of androgen actions in pubertal
 maturation of the testis, focusing on the functions of AR transactivation and transrepression. 

The manuscript was well organized and contained important findings in this field.

It might be interesting to summarize about the roles of coactivator and/or corepressor of AR in sertoli cells.

Author Response

We appreciate your input and we have addressed this issue by including a paragraph addressing co-effectors and co-repressors of AR function (lines 192-203 of the revised version).

Reviewer 2 Report

In the review entitled “Importance of the androgen receptor signalling in gene transactivation and transrepression for pubertal maturation of the testis”, the authors give an exhaustive overview of the role of  androgens in Sertoli cell tight junction formation and maintenance and,   on its effect on proper germ cell differentiation and meiotic onset during spermatogenesis.

The manuscript is structured appropriately and is written very well, detailing the topics covered.

The images presented are very clear and the contents very important.

We believe that the manuscript adds an important contribution to the understanding of Androgens roles.

Author Response

Thank you for your kind comments.

Reviewer 3 Report

Type of manuscript: Review

Title: Importance of the androgen receptor signaling in gene transactivation

and transrepression for pubertal maturation of the testis

This review by Edelsztein & Rey is very well written and provides useful information for several aspects of endocrine research, including relevance to hormone-dependent cancers.

The following comments/edits are suggested to make the MS ready for publication:

1-    The focus of the review is on the androgen -dependent changes that take place in the testis at puberty onset. Understanding these changes is important for AR biology and potential therapies. Besides AMH focus at puberty, the authors could also expand on the role of AMH levels and its correlation with tumors, biomarker role.

2-    Line 116: is it possible to speculate on the signals that drive AR expression in Sertoli cells? Are there ligand-independent functions of AR in pubertal testis?

3-    Line 293: the designation of Cyclin D1 as a tumor suppressor should be discussed more. It has co-repressor activity with AR but it is also amplified in cancer and works as a tumor promoter. Please clarify.

4-    AMH should be defined in the abstract.

Author Response

We thank you for constructive comments.

1. We appreciate your input and we have addressed this issue by briefly describing AMH use as a marker of androgen depletion induced by GnRH analogue in adult patients with prostate cancer and in chemotherapy-induced testicular toxicity (lines 96-101 of the revised version).

2. We have addressed this issue by briefly describing the potentiality of Sp1 as an inducing signal for AR gene expression (lines 122-128 of the revised version). Unfortunately, we have found no data on regulation of AR expression in Sertoli cells.

3. We appreciate your input. We mentioned Cyclin D1 just to exemplify androgen receptor regulated genes. We did not mean to bring attention to Cyclin D1 as a tumour suppressor. Therefore, we deleted the words “tumour suppressor genes” in the text to avoid bias (line 324 of the revised version). Since our review is focused on androgen receptor signalling and pubertal maturation of the testis, opening a discussion on cyclin D1 and tumour biology might divert the attention of the reader.

4. We have defined AMH in the abstract (line 18 of the revised version).